# Perceived attitudes toward LGBTQ+ physicians among Thai patients with psychiatric disorder: A multiregional cross-sectional study

Jarurin Pitanupong[1], Katti Sathaporn[1], Pichai Ittasakul[2], Nuntaporn Karawekpanyawong[3]*, Jaturaporn Sangkool[1], Suwannee Putthisri[2]

1 Department of Psychiatry, Faculty of Medicine, Prince of Songkla University, Songkhla, Thailand,
2 Department of Psychiatry, Faculty of Medicine, Ramathibodi Hospital, Mahidol University, Bangkok, Thailand, 3 Department of Psychiatry, Faculty of Medicine, Chiang Mai University, Chiang Mai, Thailand

* nuntaporn.karawek@cmu.ac.th

## Abstract

### Purpose

This study aims to examine attitudes toward LGBTQ+ (lesbian, gay, bisexual, transgender, queer or questioning, and more) physicians among Thai psychiatric patients and to identify associated factors across three regions of Thailand.

### Patients and methods

A cross-sectional survey was conducted from June to August 2023 in Central, Northern, and Southern Thailand. The Attitudes Toward LGBTQ+ Physician Questionnaire and a patient-doctor relationship questionnaire were utilized. Data analysis involved descriptive statistics and linear regression.

### Results

The study included 543 participants with a median age of 37 years (IQR 26–52), predominantly female (68.9%) and Buddhist (78.1%). The median score for perceived attitudes toward LGBTQ+ physicians was 86 (IQR 73.0–95.5), with significant regional variations. Most participants perceived LGBTQ+ physicians as normal (85.5%) and integral to society (94.5%). A significant portion disagreed with the idea that being an LGBTQ+ physician was sinful (85.3%) or immoral (84.0%). However, 20.3% expressed discomfort with the possibility of LGBTQ+ physicians conducting private physical examinations. Lower perceived attitudes were associated with older age and being Muslim. Conversely, higher education, female gender, having LGBTQ+ connections, and a reported gender non-conformity showed a correlation with more positive attitudes.

### Conclusion

Thai psychiatric patients generally hold positive attitudes toward LGBTQ+ physicians, though demographic factors influence these attitudes. There needs to be a particular focus

**Data Availability Statement:** All our anonymized data set files are available from the figshare database (DOIs: 10.6084/m9.figshare.27043768).

**Funding:** This study was supported by Chiang Mai University (Nuntaporn Karawekpanyawong, RG 38/2566), Thailand. The funders had no involvement in the study design, data collection, data analysis, decision to publish, or preparation of the manuscript. There was no additional external funding received for this study.

**Competing interests:** The authors have declared that no competing interests exist.

on improving attitudes regarding conducting private physical examinations to maximize patient comfort and trust.

## Introduction

In contemporary society, gender diversity is increasingly acknowledged, with individuals identifying beyond traditional gender labels [1]. Nevertheless, LGBTQ+ individuals remain part of sexual and gender minorities [2]. Consequently, within heteronormative cultures that reinforce binary gender roles, LGBTQ+ individuals may continue to encounter sexual stigma [3], including physicians who care for them. Perceptions of a physician's identity, including attitudes towards their mental image, behavior, professional experience, and the influence of media and advertisements, can be major barriers in the patient-physician relationship [4].

Establishing a positive doctor-patient relationship is crucial in the treatment of psychiatric patients [5], with gender playing a significant role. Research has shown that presumptuousness increases when the gender of physicians and patients differs, but decreases when they are the same [6]. Additionally, many physicians do not regularly address sexual orientation, attraction, or gender identity when discussing sexual histories, even with sexually active adolescents or patients presenting with depression, suicidal thoughts, or attempts [7]. Although multiple studies have explored the attitude of healthcare providers towards LGBTQ+ patients [8–13], and the attitudes of LGBTQ+ medical students [14], little information is available about patient attitudes toward LGBTQ+ physicians [15].

In Thailand, two prior studies examined the attitude of patients toward overtly effeminate male or overtly masculine female physicians, conducted in the Southern region [16, 17]. Another study showed that about 80% of psychiatric patients had positive attitudes towards LGBTQ+ physicians [18]. However, older age, lack of LGBTQ+ relatives or friends, and being Muslim were associated with lower perceived attitudes toward LGBTQ+ physicians. These studies were confined to Southern Thailand.

The cultural diversity of Thailand varies by region: the Central region is highly urbanized; the Northern region is characterized by Lanna culture and Buddhism, while animism is prevalent among hill tribes; the Southern region, with a significant Muslim population, adds to the country's cultural and religious diversity [19]. These cultural differences may influence attitudes toward gender minorities, with regional variations likely.

This study aimed to determine perceived attitudes toward LGBTQ+ physicians and associated factors among patients with psychiatric disorders from three regions of Thailand. It also aimed to compare these attitudes among patients with psychiatric disorders across these regions. The patient-doctor relationship was analyzed as an associated factor. The information accrued will be valuable in promoting relationships between LGBTQ+ physicians and patients with psychiatric disorders.

## Material and methods

### Respondents and procedure

After receiving approval from the ethics committees of the Faculty of Medicine at Prince of Songkla University (REC.66-001-3-1), Mahidol University (REC: MURA 2023/314), and Chiang Mai University (REC:121/2566), this cross-sectional study was conducted at three psychiatric outpatient clinics in Thailand: 1) Songklanagarind Hospital, Prince of Songkla University (Southern region), 2) Ramathibodi Hospital, Mahidol University (Central region), and 3)

Maharaj Nakorn Chiang Mai Hospital, Chiang Mai University (Northern region). Due to data collection limitations, this study did not collect information from the Western and Eastern regions. The inclusion criteria were patients aged at least 18 years with a psychiatric disorder who could read and complete the questionnaires. Patients lacking the mental capacity to complete the questionnaires, such as those with severe withdrawal, intoxication, psychotic symptoms, mania, or depression, were excluded.

The sample size was calculated based on a prior study where 14.9% of patients found overtly effeminate males or overtly masculine females unacceptable as doctors in Thai society [17], so this study used P as equal to 14.9% to calculate the sample size from the following formula.

$$n = \frac{Z_{\frac{\alpha}{2}}^2 PQ}{d^2}$$

d = 0.03 (approximately 20.0% of p), Zα/2 represented the critical value of the normal distribution at α/2 (e.g., for a confidence level of 95.0%, α was 0.05, and the critical value was 1.96), d was the margin of error, and p was the proportion of patients who acknowledged that being an overtly effeminate male or an overtly masculine female doctor was unacceptable in the Thai society. The required sample size was at least 541 patients. One hundred and eighty-one patients were required from each hospital ensuring the demographic distribution.

From June 1 to August 31, 2023, data was collected using a paper-based self-reporting process. Research assistants approached eligible patients at psychiatric outpatient clinics, explained the rationale of the study, and allowed them 20–30 minutes to decide on participation. Those who agreed signed informed consent forms, received the questionnaires, and were invited into a private space to complete the survey. Research assistants monitored participants and informed them that they could withdraw from the study at any time if they felt uncomfortable or distressed.

## Questionnaire

**1. General demographic information.** Participants provided information on age, gender, sex, religion, marital status, education, income, occupation, psychiatric diagnosis, and the presence of LGBTQ+ relatives or friends. They also shared their experiences with LGBTQ+ acquaintances and physicians, including any previous experience of being medically examined by LGBTQ+ physicians. In this study, LGBTQ+ physicians were defined as those with diverse sexual orientations and gender identities.

**2. Questionnaire investigating attitudes toward LGBTQ+ physicians.** Adapted from the Thai version of the Attitudes Toward Transgendered Individuals Scale (ATTI), this self-rating questionnaire consisted of 20 questions assessing cognitive and affective reactions to LGBTQ+ physicians. Responses were on a 5-point Likert scale, from 1 (strongly agree) to 5 (strongly disagree), with nine items reverse-scored. Total scores ranged from 20 to 100, with higher scores indicating more positive attitudes [20]. The questionnaire showed high internal consistency reliability with a Cronbach's alpha of 0.96 [21].

**3. Comparison of attitudes toward LGBTQ+ and non-LGBTQ+ physicians.** This self-rating questionnaire compared the attitudes of patients towards LGBTQ+ and non-LGBTQ+ physicians across six hospital management processes: general history-taking, private issue history-taking (e.g., sexual relations and psychiatric illness), general physical examination, private physical examination (e.g., breasts and genitalia), management of medical conditions, and management of psychiatric conditions. Each question was scored from 1 (uncomfortable) to 3 (comfortable) [16–18, 22].

**4. Patient-doctor relationship questionnaire (PDRQ-9) Thai version.** This self-rating questionnaire, adapted from the English version with a content validity index (CVI) score of

0.8 [23], included nine questions rated on a 5-point scale from 1 (not at all appropriate) to 5 (totally appropriate). Higher scores indicated a better patient-doctor relationship, with a total score range of 9 to 45; 36 or higher indicated a good relationship, 18 to 35 moderate, and 17 or lower poor [24]. The PDRQ-9 demonstrated internal consistency with a Cronbach's alpha of 0.70–0.94 [25, 26].

## Statistical analysis

Descriptive statistics were used to summarize the data, including proportions, means, standard deviations (SD), medians, and interquartile ranges (IQR). Associations between demographic characteristics and perceived attitudes toward LGBTQ+ physicians were assessed using the Wilcoxon rank-sum test, Kruskal-Wallis test, and linear regression. Relevant covariates (e.g., gender conformity, age, marital status, religion, education, occupation, diagnosis, LGBTQ+ relatives/friends, prior experience with LGBTQ+ physicians, and patient-doctor relationship) were included to control the impact of confounding variables in the multivariable regression. Variables with p < 0.200 from univariate analysis were selected. The Kruskal-Wallis test was employed to compare attitudes across regions. All analyses were conducted using R version 4.3.1, with statistical significance defined as p < 0.05.

## Results

### Demographic characteristics

From June to August 2023, a total of 543 participants attended three psychiatric outpatient clinics. The median age of the study population was 37 years (IQR: 26–52 years). The majority of participants were female (68.9%), single (58.9%), Buddhist (78.1%), and had a bachelor's degree or higher (70.5%). Gender non-conformity was reported by 29 participants (5.3%). The most common psychiatric disorders were major depressive disorder (MDD) (46.0%) and generalized anxiety disorder (GAD) (16.9%). According to the results of the PDRQ-9 questionnaire, most participants (77.3%) reported a good doctor-patient relationship. Additionally, 58.9% had LGBTQ+ close relatives or friends, and 30.8% had previous experience of being medically examined by LGBTQ+ physicians (Tables 1 and 2). A statistically significant difference in demographic data was observed among participants from the three hospitals, except for patient-doctor relationships.

When comparing the three regions, Southern participants were the most elderly, had the highest rate of ever being married, the largest proportion practicing Islam, and the highest incidence of an education level of secondary school or below. In contrast, the Northern region had the most participants with LGBTQ+ close relatives or friends and the highest number with prior experience of being medically examined by LGBTQ+ physicians.

### Attitudes toward LGBTQ physicians

Regarding attitudes toward LGBTQ+ physicians, the median (IQR) score was 86 (73.0, 95.5). The average scores among participants from the Northern, Central, and Southern regions were 93.0, 87.0, and 75.0, respectively, with the Southern region scoring significantly lower than the other two (p < 0.001). Pairwise comparisons using the Wilcoxon rank sum test showed significant differences between the Southern and Northern regions (p < 0.001), Southern and Central regions (p < 0.001), and Northern and Central regions (p = 0.029). The younger age group had significantly higher average scores than the older age group (p < 0.001).

Most participants reported positive attitudes toward LGBTQ+ physicians, with 85.5% agreeing/strongly agreeing that recognition of LGBTQ+ physicians was normal, 94.5% seeing

**Table 1. Baseline characteristics.**

| | | Total (%) (N = 543) | Northern region (%) (n = 181) | Central region (%) (n = 181) | Southern region (%) (n = 181) |
|---|---|---|---|---|---|
| Gender conformity | Gender conformity | 514 (94.7) | 171 (94.5) | 170 (93.9) | 173 (95.6) |
| | • Male | 160 (29.5) | 57 (31.5) | 34 (18.8) | 69 (38.1) |
| | • Female | 354 (65.2) | 114 (63.0) | 136 (75.1) | 104 (57.5) |
| | Gender non-conformity | 29 (5.3) | 10 (5.5) | 11 (6.1) | 8 (4.4) |
| Age (year) | 18–25 | 130 (23.9) | 65 (35.9) | 34 (19.0) | 31 (17.1) |
| | 26–40 | 181 (33.3) | 59 (32.6) | 74 (41.3) | 48 (26.5) |
| | 41–60 | 159 (29.3) | 43 (23.8) | 49 (27.4) | 67 (37.0) |
| | >60 | 71 (13.1) | 14 (7.7) | 22 (12.3) | 35 (19.3) |
| Marital Status | Single | 320 (58.9) | 120 (66.3) | 119 (65.7) | 81 (44.8) |
| | Married | 172 (31.7) | 49 (27.1) | 46 (25.4) | 77 (42.5) |
| | Divorced/widowed/ separated | 51 (9.4) | 12 (6.6) | 16 (8.8) | 23 (12.7) |
| Religion | Buddhism | 424 (78.1) | 145 (80.1) | 149 (82.3) | 130 (71.8) |
| | Islam | 45 (8.3) | 2 (1.1) | 2 (1.1) | 41 (22.7) |
| | Christianity/others | 74 (13.6) | 34 (18.8) | 30 (16.6) | 10 (5.5) |
| Education level | Secondary school or below | 52 (9.6) | 9 (5.0) | 8 (4.5) | 35 (19.3) |
| | High school/diploma | 108 (19.9) | 31 (17.1) | 30 (16.6) | 47 (26.0) |
| | Bachelor's degree or above | 383 (70.5) | 141 (77.9) | 143 (79.0) | 99 (54.7) |
| Had LGBTQ+ close relatives/friends | | 167 (30.8) | 81 (44.8) | 53 (29.4) | 33 (18.2) |

There were statistically significant differences between all variables and all three regions (p < 0.01)

them as a viable part of society, and 89.5% supporting their complete acceptance into society. Conversely, 85.3% disagreed/strongly disagreed that being an LGBTQ+ physician was a sin, and 84.0% did not see it as immoral.

According to attitudes toward LGBTQ+ physicians toward six processes of medical management, over half of the participants felt comfortable with general tasks like the taking of a

**Table 2. Characteristics related to treatment experience.**

| | | Total (%) (N = 543) | Northern region (%) (n = 181) | Central region (%) (n = 181) | Southern region (%) (n = 181) |
|---|---|---|---|---|---|
| Diagnosis | MDD | 250 (46.0) | 105 (58.0) | 70 (38.7) | 75 (41.4) |
| | GAD | 92 (16.9) | 26 (14.4) | 27 (14.9) | 39 (21.5) |
| | Panic disorder | 36 (6.6) | 15 (8.3) | 12 (6.6) | 9 (5.0) |
| | Bipolar disorder | 6 (1.1) | 14 (7.7) | 23 (12.7) | 19 (10.5) |
| | Schizophrenia | 25 (4.6) | 5 (2.8) | 4 (2.2) | 16 (8.8) |
| | Substance use | 56 (10.3) | 2 (1.1) | 0 (0.0) | 4 (2.2) |
| | Others | 78 (14.4) | 14 (7.7) | 45 (24.9) | 19 (10.5) |
| Having previous experience of being medically examined by LGBTQ+ physicians | | 167 (30.8) | 81 (44.8) | 53 (29.4) | 33 (18.2) |
| Level of patient-doctor relationship | Poor | 2 (0.4) | 1 (0.6) | 0 (0.0) | 1 (0.6) |
| | Moderate | 117 (21.5) | 49 (27.1) | 32 (18.1) | 36 (19.9) |
| | Good | 420 (77.3) | 131 (72.4) | 145 (81.9) | 144 (79.6) |

There were statistically significant differences between all variables and all three regions (p < 0.01) except for the level of patient-doctor relationship (p > 0.05).

**Table 3. Attitudes toward LGBTQ+ physicians toward six processes of medical management (N = 543).**

| Processes of medical management | Uncomfortable n (%) | Neutral n (%) | Comfortable n (%) | No answer n (%) |
|---|---|---|---|---|
| Taking a history of general issues | 9 (1.7) | 114 (21.0) | 418 (77.0) | 2 (0.4) |
| Taking a history of private issues such as sexual relations and psychiatric illness | 32 (5.9) | 147 (27.1) | 362 (66.7) | 2 (0.4) |
| Physical examination of general areas | 20 (3.7) | 148 (27.3) | 371 (68.3) | 4 (0.7) |
| Physical examination of private parts such as breasts and genitalia | 110 (20.3) | 172 (31.7) | 258 (47.5) | 3 (0.6) |
| Management of medical conditions | 13 (2.4) | 109 (20.1) | 419 (77.2) | 2 (0.4) |
| Management of psychiatric conditions | 14 (2.6) | 110 (20.3) | 417 (76.8) | 2 (0.4) |

medical history (77.0%), general physical examinations (68.3%), and managing medical or psychiatric conditions (77.2% and 76.8%, respectively). However, comfort levels were lower during history taking involving private matters such as sexual relations and psychiatric illness (66.7%) and physical examinations of private body parts, including breasts and genitalia (47.5%), by LGBTQ+ physicians (Table 3).

## Associations between demographic characteristics and perceived attitudes toward LGBTQ+ physicians

The selection process began with a univariate analysis of each variable. Variables with a significant univariate test ($p < 0.200$) were chosen as candidates for multivariate analysis. To take into account confounding variables, we performed a multivariable regression analysis, including such variables as covariates in the model (e.g., gender conformity, age group, marital status, religion, educational level, occupation, diagnosis, presence of LGBTQ+ close relatives/friends, prior experience with LGBTQ+ physicians, and patient-doctor relationship). Additionally, we conducted model diagnostics to assess multi-collinearity among the independent variables. The Variance Inflation Factor (VIF) was calculated for each variable, and all VIF values were below 5 (the commonly accepted threshold being 10), indicating that multi-collinearity was not a concern in this model. The final linear regression model revealed that age, sex, marital status, religion, education level, presence of LGBTQ+ close relatives/friends, prior experience with LGBTQ+ physicians, and patient-doctor relationship were significantly associated with perceived attitudes toward LGBTQ+ physicians.

The findings indicated that older participants had lower scores of perceived attitudes toward LGBTQ+ physicians compared to younger participants. Muslims presented as having lower scores than Buddhists (adjusted coefficient: -10.3, 95% CI = -13.3, -7.3). Participants without LGBTQ+ close relatives/friends and those with no previous experience of being examined by LGBTQ+ physicians recorded lower scores (adjusted coefficients: -6.4, 95% CI = -8.3, -4.5, and -2.8, 95% CI = -4.7, -0.89, respectively). A higher education level was positively associated with attitudes (adjusted coefficient: 5.9, 95% CI = 2.9, 8.9 for bachelor's degree or higher). Females and gender non-conforming individuals reported more positive attitudes (adjusted coefficient: 3.5, 95% CI = 1.6, 5.4 for females and 6.1, 95% CI = 2.0, 10.2 for gender non-conformity) (Table 4).

When analyzing factors associated with the Perceived Attitudes Score using Linear Regression Models by region, it was found that only age and the presence of LGBTQ+ close relatives or friends were significantly associated across all three regions. Some factors showed an association with the Perceived Attitudes Score in specific regions but not in others, including having previous experience with LGBTQ+ physicians in the Northern region, marital status in the Central region, and religion and education in the Southern region. (Table 5).

**Table 4. The linear regression models of the perceived attitudes score (N = 536).**

| Variables | Crude coefficient (95% CI) | Adjusted coefficient (95% CI) | p-value (F-test) |
|---|---|---|---|
| **Age group (year)** | | | <0.001 |
| 18-25 | Reference | Reference | |
| 26-40 | -4.36 (-7.01,-1.71) | -4.42 (-6.74,-2.09) | |
| 41-60 | -12.79 (-15.52,-10.06) | -9.23 (-11.82,-6.65) | |
| >60 | -17.81 (-21.20,-14.42) | -12.35 (-15.69,-9.00) | |
| **Marital Status** | | | 0.042 |
| Single/divorced/widow/separate | Reference | Reference | |
| Married | -9.27 (-11.55,-6.98) | -2.12 (-4.17,-0.08) | |
| **Religion** | | | <0.001 |
| Buddhism | Reference | Reference | |
| Islam | -10.31 (-14.27,-6.35) | -10.3 (-13.34,-7.26) | |
| Christianity/ others | 3.88 (0.66,7.10) | -0.26 (-2.80,2.27) | |
| **Education** | | | <0.001 |
| Secondary school and below | Reference | Reference | |
| High school/diploma | 6.22 (2.06,10.38) | 1.87 (-1.45,5.20) | |
| Bachelor's degree and above | 13.07 (9.43,16.71) | 5.94 (2.91,8.98) | |
| **Sex** | | | <0.001 |
| Male | Reference | Reference | |
| Female | 6.09 (3.70,8.48) | 3.51 (1.60,5.42) | |
| Gender non-conformity | 15.70 (10.66,20.74) | 6.13 (2.03,10.24) | |
| **Had LGBTQ+ close relatives/friends** | | | <0.001 |
| Yes | Reference | Reference | |
| No | -12.15 (-14.19,-10.11) | -6.40 (-8.29,-4.51) | |
| **Having previous experience of being medically examined by LGBTQ+ physicians** | | | 0.004 |
| Yes | Reference | Reference | |
| No | -7.65 (-10.00,-5.31) | -2.80 (-4.70,-0.89) | |
| **Patient-doctor relationship** | | | <0.001 |
| Moderate | Reference | Reference | |
| Good | 3.28 (0.59,5.97) | 5.09 (3.04,7.14) | |

## Discussion

To our knowledge, this is the first multiregional hospital-based survey investigating the perceived attitudes toward LGBTQ+ physicians and any associating factors among patients with psychiatric disorders in Thailand. Although prior studies reported significant discrimination against LGBTQ+ individuals in Thailand [27, 28], this study found that the average score for perceived attitudes toward LGBTQ+ physicians (83.7) was higher than those recorded concerning the attitudes of US college students toward LGBTQ+ individuals (76.0) [29]. This suggests that Thai psychiatric patients hold positive attitudes toward LGBTQ+ physicians. This result may reflect a shift toward the normalization of perspectives related to LGBTQ+ [3] and the higher economic and social capital of physicians, providing them with greater protective factors against discrimination [30].

Most participants recognized LGBTQ+ physicians as normal, viable members of society, and not immoral. This reflects positive attitudes toward LGBTQ+ people among patients with psychiatric disorders, likely influenced by society's increasingly positive view of LGBTQ+ individuals [31].

**Table 5. The linear regression models of the perceived attitudes score by the three region.**

| Variables | Northern region | | Central region | | Southern region | |
|---|---|---|---|---|---|---|
| | Adjusted coefficient (95% CI) | p-value (F-test) | Adjusted coefficient (95% CI) | p-value (F-test) | Adjusted coefficient (95% CI) | p-value (F-test) |
| **Age group (year)** | | <0.001 | | <0.001 | | <0.001 |
| 18-25 | Reference | | Reference | | Reference | |
| 26-40 | -5.76 (-8.59,-2.92) | | 1.37 (-2.65,5.39) | | -7.99 (-13.26,-2.71) | |
| 41-60 | -11.48 (-14.62,-8.34) | | -3.75 (-8.30,0.81) | | -12.61 (-17.75,-7.47) | |
| >60 | -15.61 (-20.58,-10.63) | | -9.61 (-15.2,-4.03) | | -12.96 (-18.98,-6.94) | |
| **Marital Status** | | - | | 0.02 | | - |
| Single/divorced/widowed/separated | - | | Reference | | - | |
| Married | - | | -4.16 (-7.66,-0.66) | | - | |
| **Religion** | | - | | - | | 0.001 |
| Buddhism | - | | - | | Reference | |
| Islam | - | | - | | -7.95 (-12.13,-3.76) | |
| Christianity/others | - | | - | | -2.41 (-10.13,5.31) | |
| **Education** | | - | | - | | 0.014 |
| Secondary school and below | - | | - | | Reference | |
| High school/diploma | - | | - | | 1.63 (-3.49,6.74) | |
| Bachelor's degree and above | - | | - | | 6.26 (1.55,10.97) | |
| **Sex** | | 0.002 | | 0.001 | | - |
| Male | Reference | | Reference | | - | |
| Female | 4.70 (2.14,7.27) | | 5.67 (1.89,9.45) | | - | |
| Gender non-conformity | 3.17 (-2.27,8.61) | | 11.23 (4.45,18.01) | | - | |
| **Had LGBTQ+ close relatives/friends** | | <0.001 | | <0.001 | | <0.001 |
| Yes | Reference | | Reference | | Reference | |
| No | -4.87 (-7.68,-2.06) | | -5.75 (-8.88,-2.61) | | -7.68 (-11.24,-4.11) | |
| **Having previous experience of being medically examined by LGBTQ+ physicians** | | 0.006 | | - | | - |
| Yes | Reference | | - | | - | |
| No | -3.23 (-5.54,-0.92) | | - | | - | |
| **Patient-doctor relationship** | | <0.001 | | 0.001 | | - |
| Moderate[a] | Reference | | Reference | | - | |
| Good | 6.60 (4.02,9.17) | | 6.30 (2.57,10.03) | | - | |

[a] Including 1 case reporting poor relationship in Northern region.

LGBTQ+, Lesbian, gay, bisexual, transgendered, queer, and more; ATTI, The Attitudes Toward Transgendered Individuals Scale; PDRQ, A patient-doctor relationship questionnaire.

Regarding attitudes toward LGBTQ+ physicians across six medical management processes, most participants felt comfortable with LGBTQ+ physicians during general history-taking. However, they were less comfortable when discussing sensitive matters such as sexual relations and psychiatric illness. While over half of the participants reported feeling at ease with LGBTQ + physicians conducting general physical examinations, less than half were comfortable with examinations involving private body areas, such as the breasts and genitalia. Nevertheless, most patients remained comfortable with LGBTQ+ physicians managing either their medical or psychiatric conditions.

This information is important for LGBTQ+ professionals in increasing their awareness regarding their patients' feelings to ensure the maintenance of trustworthy care, particularly in psychiatric settings where a higher degree of trust is essential. Psychiatric patients often

struggle to form relationships and may be more susceptible to mistrust [32]. It's important to note that "LGBTQ+" covers a wide range of identities, and patients may feel discomfort if examined by doctors of the opposite sex, regardless of the doctors' LGBTQ+ status [33]. For example, female patients may feel more comfortable with a trans-feminine physician than a heterosexual cisgender male physician but are most comfortable with a cisgender female physician. Other psychological factors may also contribute to this discomfort. Discomfort during private examinations might be similar among patients with general medical conditions or the public, warranting further study.

Despite the generally high attitude scores, significant regional differences were observed, with the Northern and Central regions reporting higher scores than the Southern region. This variation may be attributed to differences in participant characteristics. For instance, the sample from the Southern region had an older age demographic and a higher proportion of Muslim participants compared to other regions. Previous studies have shown that older age and being Muslim are associated with less positive attitudes toward LGBTQ+ physicians [16, 17]. In contrast, participants from the Northern region reported more experience having LGBTQ+ close relatives or friends and being treated by LGBTQ+ physicians. Both this study and previous research [14, 34] found that having LGBTQ+ close relatives or friends is associated with more positive attitudes toward LGBTQ+ individuals. These findings highlight the importance of personal contact and relationships in shaping attitudes and may explain the association between positive attitudes and prior treatment by LGBTQ+ physicians. Other factors associated with higher attitude scores included being female, reported gender non-conformity, and having a higher level of education, consistent with prior studies in Thailand.

The findings from this study can inform the development of strategies to foster positive attitudes toward LGBTQ+ individuals, potentially by focusing on specific areas in the future. For instance, enhancing understanding and perceptions through education and providing patients with opportunities for positive interactions with LGBTQ+ individuals or physicians in a safe environment [35], particularly in the case of older adults and Muslim patients, could be beneficial. To ensure optimal care, LGBTQ+ physicians should focus on building trust with patients, especially those from particular demographic groups. This process could begin with medical procedures that patients are typically more comfortable with, such as general history taking, and gradually introduce more sensitive procedures, such as psychiatric and sexual histories, while exercising caution during physical examinations of private areas, such as the breasts and genitalia. Additionally, it is important to regularly assess patients' feelings and comfort levels throughout the treatment process.

This study had a few noteworthy strengths and limitations. To our knowledge, this was the first study investigating the perceived attitudes toward LGBTQ+ physicians among patients with psychiatric disorders in Thailand. This provided a current view of the perceived attitudes of psychiatric patients toward LGBTQ+ physicians. The sample size consisted of patients enrolled from various regions in multiregional Thailand, each with different characteristics, encompassing all regions. However, no data was collected from two regions, the Eastern and Western regions. These two regions may present with characteristics that are either similar or different from the rest. Therefore, caution is advised when generalizing these results. In addition, the study's findings are limited by its cross-sectional design, which restricts the scope of interpretation. Also, the use of self-administered questionnaires may have led to potential misunderstandings of the intended meanings of the questions. Other drawbacks were that the data were quantitative, and the questionnaires covered the attitudes towards LGBTQ+ physicians as a whole without distinguishing between attitudes toward LGBTQ+ and non-LGBTQ+ physicians, resulting in no in-depth comparison of the differences in attitudes between the two groups of physicians. Future studies might compare the attitudes of patients with

psychiatric disorders towards all physicians, covering both groups about sexual orientation and other issues related to gender identities. Additionally, most participants were patients with MDD, and GAD conditions which do not cover all psychiatric disorders; hence, this dataset might not fairly represent Thai patients with psychiatric disorders countrywide. Therefore, future studies should include patients with all psychiatric disorders and should employ different instruments, utilizing more qualitative designs, or in-depth studies.

## Conclusion

Most participants reported positive attitudes toward LGBTQ+ physicians, but there were differences in scores between the three regions. Some age groups or religions reported lower scores regarding attitudes toward LGBTQ+ physicians. There is therefore potential for concern that needs addressing. Positive attitudes toward LGBTQ+ physicians must be promoted among these groups of people.

## Acknowledgments

The authors would like to thank the participants for their willingness to provide information. We also wish to acknowledge the assistance of Associate Professor Hutcha Sriplung and the research assistants, Nisan Werachattawan and Kruewan Jongborwanwiwat, for their support. The English in this article was proofread and edited by the Office of International Affairs, Faculty of Medicine, Prince of Songkla University, and the Research Administration Section, Faculty of Medicine, Chiang Mai University.

## Author Contributions

**Conceptualization:** Katti Sathaporn.

**Data curation:** Jarurin Pitanupong, Katti Sathaporn, Pichai Ittasakul, Nuntaporn Karawekpanyawong, Jaturaporn Sangkool, Suwannee Putthisri.

**Formal analysis:** Jarurin Pitanupong, Katti Sathaporn, Pichai Ittasakul, Nuntaporn Karawekpanyawong, Jaturaporn Sangkool, Suwannee Putthisri.

**Funding acquisition:** Nuntaporn Karawekpanyawong.

**Visualization:** Nuntaporn Karawekpanyawong.

**Writing – original draft:** Jarurin Pitanupong, Katti Sathaporn.

**Writing – review & editing:** Jarurin Pitanupong, Katti Sathaporn, Pichai Ittasakul, Nuntaporn Karawekpanyawong, Jaturaporn Sangkool, Suwannee Putthisri.

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
