## [Decision Letter · Decision Letter 0]

6 Sep 2024

PONE-D-24-32784Perceived attitudes toward LGBTQ+ physicians among Thai patients with psychiatric disorder: a multiregional cross-sectional studyPLOS ONE

Dear Dr. Karawekpanyawong,

Thank you for submitting your manuscript to PLOS ONE. After careful consideration, we feel that it has merit but does not fully meet PLOS ONE’s publication criteria as it currently stands. Therefore, we invite you to submit a revised version of the manuscript that addresses the points raised during the review process.

 As you can see, the reviewers are overall excited about this research as am I. Please ensure to also address my additional comments below that go beyond the points mentioned by the reviewers. 

We look forward to receiving your revised manuscript.

Kind regards,

Daniel Demant, PhD, MPH, GradCertHEd, BAppSocSc

Academic Editor

PLOS ONE

Journal Requirements:

"This study was supported by Chiang Mai University (Nuntaporn Karawekpanyawong, RG 38/2566), Thailand. The funders had no involvement in the study design, data collection, data analysis, decision to publish, or preparation of the manuscript."

Additional Editor Comments:

Grammar and Language

The manuscript contains a number grammatical errors and awkward sentence structures that affect readability. Inconsistent use of tense throughout the manuscript makes it difficult to follow the narrative. The flow between some sections, specifically in the introduction is sometimes poor. I would recommend a comprehensive language edit by a professional or a colleague who speaks English as a first language.

Methodology

The study employs a cross-sectional design, which is inherently limited in establishing causality. This limitation should be more explicitly discussed in the methodology and discussion sections. There is no justification provided for the selection of the three specific regions of Thailand, which limits the generalisability of the findings. The statistical methods used are appropriate, but the manuscript lacks a clear explanation of how confounding variables were controlled in the regression analysis.

Clarity of Results

The results section is cluttered with excessive numerical data without adequate contextual interpretation. Tables and figures are used but are not sufficiently explained in the text. A brief summary of the key points from each table should be included to aid reader comprehension.

Critical Level of the Discussion

The discussion provides a surface-level interpretation of results without delving deeply into potential reasons behind the observed patterns. There is an over-reliance on describing findings rather than critically engaging with them. Suggestions for future research are made but lack specificity and do not fully address the study's limitations. As an example, the differences in attitudes between the regions are reported, but there is little exploration of why these differences might exist beyond brief mentions in the discussion. Comparisons with existing literature are limited, reducing the critical engagement with the field. A more extensive literature integration would strengthen the discussion.

There is an over-reliance on describing findings rather than critically engaging with them. Suggestions for future research are made but lack specificity and do not fully address the study's limitations.

Reviewers' comments:

Reviewer's Responses to Questions

**Comments to the Author**

1. Is the manuscript technically sound, and do the data support the conclusions?

Reviewer #1: Yes

Reviewer #2: Yes

2. Has the statistical analysis been performed appropriately and rigorously? 

Reviewer #1: Yes

Reviewer #2: Yes

3. Have the authors made all data underlying the findings in their manuscript fully available?

Reviewer #1: Yes

Reviewer #2: Yes

4. Is the manuscript presented in an intelligible fashion and written in standard English?

Reviewer #1: Yes

Reviewer #2: Yes

5. Review Comments to the Author

Reviewer #1: I would like to congratulate the authors of the article for bringing such an interesting point of view in the diversity discussion.

I think it would help the article if the authors put the "p value" in table 3, in order to help the comparisons.

Reviewer #2: �Overall, the study is well-conducted and informative. The findings on perceived attitude toward LGBTQ+ physicians among Thai patients with psychiatric disorder, are supported by clear statistical results and are appropriately discussed within the context of current literature.

6. PLOS authors have the option to publish the peer review history of their article (what does this mean?). If published, this will include your full peer review and any attached files.

Reviewer #1: **Yes: **I would like to congratulate the authors of the article for bringing such an interesting point of view in the diversity discussion.

I think it would help the article if the authors put the "p value" in table 3, in order to help the comparisons.

Reviewer #2: No

---

## [Author Response · Author response to Decision Letter 0]

17 Sep 2024

Response to reviewer

 We would like to thank you for your careful consideration of our manuscript, your pertinent questions, and your helpful suggestions. Your comments have led to several important changes to our manuscript that have considerably improved it in our opinion. We have revised our manuscript according to your recommendation, as follows:

The comments by the reviewers appear in the boxes below. The response to these comments and the associated change to the manuscript are listed below. The changes are also highlighted in the revised manuscript.

Answer:

We have revised the manuscript to ensure it adheres to PLOS ONE's style requirements.

"This study was supported by Chiang Mai University (Nuntaporn Karawekpanyawong, RG 38/2566), Thailand. The funders had no involvement in the study design, data collection, data analysis, decision to publish, or preparation of the manuscript."

Answer: 

We have revised the funding statement as follows:

"This study was supported by Chiang Mai University (Nuntaporn Karawekpanyawong, RG 38/2566), Thailand. The funders had no involvement in the study design, data collection, data analysis, decision to publish, or preparation of the manuscript. There was no additional external funding received for this study."

Additional Editor Comments:

Grammar and Language

The manuscript contains a number grammatical errors and awkward sentence structures that affect readability. Inconsistent use of tense throughout the manuscript makes it difficult to follow the narrative. The flow between some sections, specifically in the introduction is sometimes poor. I would recommend a comprehensive language edit by a professional or a colleague who speaks English as a first language.

Answer:

We have thoroughly edited the manuscript to improve clarity in English usage.

Methodology

The study employs a cross-sectional design, which is inherently limited in establishing causality. This limitation should be more explicitly discussed in the methodology and discussion sections. There is no justification provided for the selection of the three specific regions of Thailand, which limits the generalisability of the findings. The statistical methods used are appropriate, but the manuscript lacks a clear explanation of how confounding variables were controlled in the regression analysis.

Answer:

- We have noted the reasons for selecting the three specific regions of Thailand due to the limitation of data in the Methods section. Please see page 5, lines 88-89.

- We have discussed the limitation of generalization of the finding due to region selection in the discussion section. Please see page 18, lines 309-311.

- We have explained how confounding variables were controlled in the regression analysis in the Methods section. Please see pages 7-8, lines 145-149.

Clarity of Results

The results section is cluttered with excessive numerical data without adequate contextual interpretation. Tables and figures are used but are not sufficiently explained in the text. A brief summary of the key points from each table should be included to aid reader comprehension.

Answer: 

- We have summarized the key points from the Results section to aid reader comprehension. For Tables 1 and 2, please refer to pages 8-9, lines 156-169. 

For Table 3, please refer to page 11, lines 192-198. 

For Table 4, please refer to pages 12-13, lines 218-227. 

For Table 5, please refer to page 14, lines 231-236.

Critical Level of the Discussion

The discussion provides a surface-level interpretation of results without delving deeply into potential reasons behind the observed patterns. There is an over-reliance on describing findings rather than critically engaging with them. Suggestions for future research are made but lack specificity and do not fully address the study's limitations. As an example, the differences in attitudes between the regions are reported, but there is little exploration of why these differences might exist beyond brief mentions in the discussion. Comparisons with existing literature are limited, reducing the critical engagement with the field. A more extensive literature integration would strengthen the discussion.

Answer: 

- We have provided a deeper discussion of potential reasons behind the results, such as differences in attitudes between regions. Please refer to page 17, lines 278-291.

- We have suggested specific directions for future research. Please refer to pages 18-19, lines 322-324.

- We have fully addressed the study's limitations. Please refer to pages 18-19, lines 304-324.

Reviewer #1: 

I would like to congratulate the authors of the article for bringing such an interesting point of view in the diversity discussion.

I think it would help the article if the authors put the "p value" in table 3, in order to help the comparisons.

Answer: 

We apologize for any misunderstanding. Table 3 presents the attitudes toward LGBTQ+ physicians across six processes of medical management for all participants, without comparisons between processes or between attitudes toward non-LGBTQ+ physicians for these processes. Therefore, no p-values are provided for this result.

---

## [Editor Report · Decision Letter 1]

25 Sep 2024

Perceived attitudes toward LGBTQ+ physicians among Thai patients with psychiatric disorder: a multiregional cross-sectional study

PONE-D-24-32784R1

Dear Dr. Karawekpanyawong,

We’re pleased to inform you that your manuscript has been judged scientifically suitable for publication and will be formally accepted for publication once it meets all outstanding technical requirements.

Kind regards,

Daniel Demant, PhD, MPH, GradCertHEd, BAppSocSc

Academic Editor

PLOS ONE
---

## [Editor Report · Acceptance letter]

29 Oct 2024

PONE-D-24-32784R1 

PLOS ONE

Dear Dr. Karawekpanyawong, 

I'm pleased to inform you that your manuscript has been deemed suitable for publication in PLOS ONE. Congratulations! Your manuscript is now being handed over to our production team.

Kind regards, 

on behalf of

Dr. Daniel Demant 

Academic Editor

PLOS ONE